# The Usefulness of Intraepithelial Lymphocyte Immunophenotype Testing for the Diagnosis of Coeliac Disease in Clinical Practice

**DOI:** 10.3390/nu16111633

**Published:** 2024-05-26

**Authors:** Laura Gutiérrez-Rios, Margalida Calafat, Irene Pascual, Cristina Roig, Aina Teniente-Serra, Laia Vergés, Carlos González-Muñoza, Eva Vayreda, Diego Vázquez, Jordi Gordillo, Míriam Mañosa, Consuelo Ramírez, Esther Garcia-Planella, Montserrat Planella, Eugeni Domènech

**Affiliations:** 1Gastroenterology Department, Hospital Universitari Germans Trias i Pujol, 08029 Badalona, Spain; lgutirrezr.germanstrias@gencat.cat (L.G.-R.); evayreda.germanstrias@gencat.cat (E.V.); mmanosa.germanstrias@gencat.cat (M.M.); edomenech.germanstrias@gencat.cat (E.D.); 2Centro de Investigaciones Biomédicas en Red de Enfermedades Hepáticas y Digestivas (CIBEREHD), 28029 Madrid, Spain; 3Gastroenterology Department, Hospital Universitari Arnau de Vilanova, 25198 Lleida, Spain; ipascuall.lleida.ics@gencat.cat (I.P.); lverges.lleida.ics@gencat.cat (L.V.); dvazquezg.lleida.ics@gencat.cat (D.V.); cramirez.lleida.ics@gencat.cat (C.R.); mplanella.lleida.ics@gencat.cat (M.P.); 4Gastroenterology Department, Hospital de la Santa Creu i Sant Pau, 08041 Barcelona, Spain; croigr@santpau.cat (C.R.); cgonzalezm@santpau.cat (C.G.-M.); egarciapl@santpau.cat (E.G.-P.); 5Immunology Department, Hospital Universitari Germans Trias i Pujol, 08029 Badalona, Spain; ateniente.germanstrias@gencat.cat; 6Digestive Diseases Research Group (DdRG)-IRBLleida, 25198 Lleida, Spain; 7Medicine School, Universitat Autònoma de Barcelona, 08193 Barcelona, Spain

**Keywords:** coeliac disease, gluten-free diet, immunophenotype, intraepithelial lymphocytes

## Abstract

Background: The diagnosis of coeliac disease (CD) in adults is based on clinical, serological and histological criteria. The inappropriate performance of intestinal biopsies, non-specificity of mild histological lesions and initiation of a gluten-free diet (GFD) before biopsy may hamper the diagnosis. In these situations, determining the intraepithelial lymphogram of the duodenum by flow cytometry (IEL-FC) can be helpful. Objectives: To describe the clinical scenarios in which the IEL-FC is used and its impact on the diagnosis of CD. Methods: All adult patients with suspected CD at three tertiary centres for whom the duodenal histology and IEL-FC were available were identified. Catassi and Fasano’s diagnostic criteria and changes to a CD diagnosis after the IEL-FCs were collected. Results: A total of 348 patients were included. The following indications for an IEL-FC formed part of the initial study for CD (38%): negative conventional work-up (32%), already on a GFD before duodenal biopsies (29%) and refractoriness to a GFD (2%). The IEL-FC facilitated a definitive diagnosis in 93% of patients with an uncertain diagnosis who had had a conventional work-up for CD or who were on a GFD before histology. Conclusions: The IEL-FC facilitates the confirmation or rejection of a diagnosis of CD in clinical scenarios in which a conventional work-up may be insufficient.

## 1. Introduction

Coeliac disease (CD) is an immune-mediated systemic disorder elicited by dietary gluten intake in genetically predisposed individuals. It results mainly in injury to the small bowel, although it has a wide spectrum of clinical manifestations and thereby resembles a multisystemic disorder rather than an isolated intestinal disease [1,2,3]. Treatment of CD requires strict adherence to a lifelong gluten-free diet (GFD), which has important consequences not only regarding dietary habits but also for patients’ social lives, as well as being an economic burden. An unequivocal diagnosis is, therefore, of vital importance.

A CD diagnosis is based on Catassi and Fasano’s criteria, which include CD-related symptoms, genetic predisposition (as shown by a positive HLA-DQ2 or HLA-DQ8), identification of serum IgA antibodies targeting tissue-transglutaminase 2 (anti-tTG), histological findings of enteropathy on duodenal biopsies and evidence of clinical and histological responses to a GFD. A CD diagnosis is reached when four out of these five criteria are fulfilled or, in the absence of a genetic study, three out of four [4]. Nevertheless, an intestinal biopsy is still required for CD diagnosis in adults.

Many patients are now diagnosed with CD while being paucisymptomatic or with quiescent disease due to an increase in CD awareness, as well as to the implementation of serological screening programmes in high-risk populations [5,6,7,8,9]. Approximately 10% of cases are difficult to diagnose because of their mild histological abnormalities or a lack of concordance between the serology and histology [2,10]. In addition, the biopsy interpretation may be hampered by sampling errors or minimal histological changes (Marsh 1), which can also be seen in other conditions such as infections, inflammatory bowel disease and non-steroidal anti-inflammatory drug-induced enteropathy [11,12,13]. Finally, some patients are already on a GFD when they are referred to the gastroenterologist for a duodenal biopsy and diagnostic confirmation. In fact, 5% to 13% of patients with CD have undergone a previous gastroduodenoscopy without biopsies or with inappropriate histological sampling, leading to a delay in diagnosis [7,8,9]. 

In light of this clinical background, new diagnostic tools have been developed in recent years, including the study of the immunophenotype of intraepithelial lymphocytes (IEL) of the duodenal mucosa by flow cytometry (IEL-FC) [5,6,7,8,9,10,11]. Patients with CD show an increased number of T-cell receptor gamma-delta+ (TCRγδ+) IEL, along with a decrease in CD3-CD103+ IEL in most cases. Two cytometric patterns have been previously described: a *complete coeliac* pattern (TCRγδ+ IEL > 8.5% and decreased CD3-CD103+ < 10%) and an *incomplete coeliac* pattern (TCRγδ+ IEL > 8.5% with normal CD3-CD103+ > 10%), both presenting with a specificity over 90% [10,14,15,16,17,18,19]. Nonetheless, a current cohort with a total of 768 adult patients has been validated, establishing a new optimal cut-off with higher diagnostic accuracy (TCRγδ+ IEL > 14% and decreased CD3-CD103+ ≤ 4%) [20]. In addition, it has been shown that TCRγδ+ IEL remains elevated in patients with CD despite a GFD [16,17]. Therefore, the IEL-FC has emerged as a useful tool when diagnosis is uncertain, decreasing both under- and overdiagnosis. However, the IEL-FC has not been included among the diagnostic criteria and its clinical usefulness is yet to be established.

We aim to describe the current use of the IEL-FC in clinical practice and assess its impact on CD diagnosis and the correlation between the cytometric pattern and serological, clinical, and histological criteria.

## 2. Methods

This is an observational, retrospective, multicentre study performed at three academic centres in Catalonia (Spain). All adults with suspected or uncertain CD for whom the duodenal IEL-FC was available were included. The study was approved by the Ethics Committee of the coordinating centre (Hospital Universitari Germans Trias i Pujol); Approval Code: PI-23-158; Approval Date: 26 March 2024.

Demographic data, concomitant diseases and the diagnostic characteristics encompassed in Catassi and Fasano’s criteria, including the clinical features of CD, serological status, HLA-genotyping (HLA-DQ2.5/2.2/8) and duodenal histology, were registered. In terms of demographic data, a high-risk group was defined by a first-degree relative history of CD or a personal history of well-established CD-associated diseases (Hashimoto’s thyroiditis, autoimmune hepatitis, Turner`s syndrome, Down’s syndrome, Sjögren’s disease, Type I diabetes mellitus or systemic lupus erythematosus). A positive serological status was defined by positive IgA anti-tTG or anti-endomysium (EmA) antibodies; in cases of IgA deficiency, positive IgG anti-tTG or EmA antibodies were needed. For those with positive anti-tTG antibodies, quantitative titres were also registered. Histological findings were described using the modified Marsh classification [2,10]. Moreover, the indication for an IEL-FC was recorded, and four clinical indication groups were encountered, namely: (a) initial CD work-up, (b) uncertain CD diagnosis, (c) already on a GFD before duodenal biopsies, and (d) refractory CD. The group with an uncertain CD diagnosis was comprised of patients with a negative serology but coeliac clinical features; Marsh 1 findings in the duodenal histology, regardless of serologic status; and patients with CD clinical features with a negative serology, unspecific histologic findings (Marsh 1 or 2) and, if available, a positive HLA-testing. We also registered whether the patient was already on a GFD at the time that the IEL-FC was performed.

The definitive diagnosis (CD confirmed, uncertain or ruled out) was given by the attending physician after the IEL immunophenotyping was registered. The IEL-FC served as confirmatory support in those patients with an initial CD work-up if they met the diagnostic criteria. In those patients who followed a GFD, patients with suspected refractory CD or uncertain diagnosis of coeliac disease, the complete IEL pattern was considered as a diagnostic criterion for CD. The incomplete pattern was interpreted as diagnostic of CD in case of a consistent clinical background of CD but with doubtful features to establish the standard diagnosis (such as a negative serology or Marsh 0–1). According to the definitive diagnosis, the proportion of patients to whom a GFD was recommended by the physician in charge was also recorded. 

For the IEL-FC, a single duodenal biopsy from both the duodenal bulb and the second portion of the duodenum was obtained at the index endoscopy. This was a part of the biopsies for the histological study. Both fresh duodenal samples were processed immediately at the immunology department at each participating centre. The samples were stored at 4 °C in a complete medium and were processed in the first hour after biopsy sampling. To remove the villous epithelium and, partially, the crypt epithelium, samples were incubated for 90 min in a solution of 1 mM EDTA and 1 mM DTT in HBSS in a vertical shaker. The cellular suspension was washed in fresh HBSS and the IELs were immediately stained with the appropriate amounts of monoclonal antibodies for 15 min at room temperature. The antibodies used were anti-CD45, anti-CD3 and anti-TCRγδ. All centres used similar gating strategies to select both CD3- and TCRγδ+ cells, which were measured as CD45+CD3-CD103+ and CD45+CD103+TCRyd+, respectively, over the total CD45+CD103+ intraepithelial cells [14,15,16,17,18]. The coeliac immunophenotype was defined by a proportion of TCRγδ+-positive cells ≥ 8.5% together with a proportion of CD3-negative cells ≤10%. An incomplete coeliac pattern was defined by an isolated increase in TCRγδ+, as previously defined in the literature [2,14,15,16,17,18].

## 3. Statistical Analysis

The quantitative variables are presented as the mean and standard deviation or median and interquartile range (IQR), depending on their distribution. The categorical variables are presented as raw numbers and proportions. The categorical variables were compared using the Chi2-squared test, and the quantitative variables using the *t*-Student test. The multivariate analysis of risk factors was assessed using logistic regression for the variables found to be statistically significant in the univariate analysis (*p* < 0.1).

## 4. Results

A total of 393 patients were identified, of whom 45 were excluded because they did not fulfil the inclusion criteria or had a negative genetic study when the IEL-FC was performed as an initial CD work-up, and 348 patients were finally included and analysed. The observed clinical features of CD are summarised in Table 1. Overall, two-thirds of patients had gastrointestinal symptoms, one-fifth presented with iron deficiency anaemia, and less than 10% were considered to be a high-risk population. The anti-tTG results were positive in 27%, and the EmA results were available for 273 patients (78%), of whom 36 (13%) were positive. Genetic testing was available for 295 patients (85%), of whom the HLA-DQ2 or HLA-DQ8 results were positive in 61%. In 179 patients (51%), a duodenal biopsy prior to the one for the IEL-FC was available, and among them, 145 patients (81%) had no or mild histological enteropathy (48% Marsh 0 and 33% Marsh 1), whereas 17% had villous atrophy (Marsh 3). Only 2% of the cohort had limited cryptal hyperplasia as defined by a modified Marsh-2.

Regarding the IEL-FC, the main indication was the initial CD work-up, which was comprised of 137 patients (38%), followed by “uncertain CD diagnosis” (112 patients, 32%) and “already on GFD before duodenal biopsies” (94 patients, 29%). Finally, suspected refractory CD was an indication for the IEL-FC in five patients (1%) (Table 2). Overall, a normal IEL-FC was found in 203 patients (58%), mainly in patients with non-definitive histology and negative serology, allowing us to rule out the diagnosis of CD. Conversely, CD diagnosis was confirmed by a complete coeliac pattern in 99 patients (28%). When only those patients with an uncertain CD diagnosis according to conventional criteria and those who were on a GFD before biopsy sampling were considered, a similar proportion of diagnoses was ruled out (64% and 55%, respectively) or confirmed (26% and 36%, respectively). Overall, following the performance of an IEL-FC and considering its various indications, physicians were able to definitively exclude CD in 62% and confirm the diagnosis in 31% of cases, respectively (Figure 1).

### Correlation between Intraepithelial Lymphocyte Immunophenotype, Histological Findings and Serological Status

A significant statistical association between the IEL-FC results and the histological findings was found (Figure 2). Almost three-fourths of patients with villous atrophy showed a complete CD pattern in the IEL-FC, whereas 73% of patients with Marsh 0 presented a normal pattern in the IEL-FC (*p* < 0.001). Similar results were observed when analysing the IEL-FC results according to the serological status. Two-thirds of the patients with anti-tTG titters > 100 had a complete CD pattern; conversely, 76% of seronegative patients showed a normal pattern in the IEL-FC (Figure 3).

Table 3 summarises the combined results of the serology, histology and IEL-FC. In fact, among patients with non-definitive histology, 13% of those with a negative anti-tGT and 50% of those with a positive anti-tGT showed a complete CD pattern at the time of the IEL-FC. On the other side, among patients with duodenal villous atrophy but a negative serology, 39% showed a complete CD pattern at the time of the IEL-FC.

To identify the potential clinical scenarios in which the performance of an IEL-FC is not useful, we performed an exploratory analysis of the correlation between clinical and serological factors and a definitive histological finding of CD (defined by Marsh 3) (Table 4). In the univariate analysis, the presence of HLA-DQ2, HLA-DQ8, anti-tTG and EmA were associated with duodenal villous atrophy in the biopsy samples; however, only anti-tTG positivity was independently associated with Marsh 3 enteropathy (*p* < 0.001). However, among those patients in whom an IEL-FC was indicated as part of the initial CD work-up despite a positive serology (n = 36), 26% had mild histological findings at the duodenal biopsy, suggesting that the IEL-FC might be useful even in this subset of patients.

## 5. Discussion

CD diagnosis is still challenging in some clinical settings. In the last decades, great efforts have been focused on seronegative patients, patients showing only mild enteropathy in duodenal biopsies and on enabling an early diagnosis in individuals already on a GFD. In these settings, the IEL-FC has been shown to provide useful information for decision-making [16,17,18]. To date, many studies have addressed diagnostic accuracy and validated the use of the IEL-FC in patients with known CD [14,15,16,17,18,19], but it has not been incorporated into the diagnostic algorithm of CD yet. In the present study, we assessed the current use of the IEL-FC in clinical practice in a large series of more than 300 patients, this being the largest cohort of patients to date, with either an established diagnosis or a high clinical suspicion of CD in which different indications for an IEL-FC were explored alongside the well-established ones (uncertain diagnosis). Among patients with an uncertain CD diagnosis or patients with suspected CD but who were already on a GFD, an IEL-FC allowed for ruling out CD in 60% of patients and confirmed the diagnosis in 31%.

Our study shows that once an IEL-FC is introduced in a centre, its utilisation spreads widely, and it is applied to any patient with suspected CD, even though an IEL-FC is likely unnecessary for those patients for whom a diagnosis could be reached simply with the “four of five” rule. Therefore, the development of a pre-test (IEL-FC) score would increase its cost-efficacy. For that reason, we performed an exploratory analysis to address which baseline features (before biopsy sampling) were associated with an unequivocal CD diagnosis (Marsh 3 enteropathy) in order to prevent unnecessary IEL-FCs. The multivariate analysis showed that a positive anti-tTG result was the only independent factor associated with the presence of villous atrophy in the biopsy samples. However, 26% of patients with baseline positive anti-tTG antibodies seemed to benefit still from an IEL-FC.

Considering the doubtful cases, the World Gastroenterology Organization guidelines for CD diagnosis recommend a second biopsy sampling in patients in whom the initial biopsies and serological tests had been inconclusive [9]. However, this strategy entails a second invasive procedure and a considerable delay in the final diagnosis. In our cohort, less than one-fifth of the patients showed villous atrophy (Marsh-modified classification 3) at the baseline duodenal biopsies. Bearing this in mind, more than 45% of the patients underwent a second gastroscopy using the IEL-FC, of whom 81% had mild histological enteropathy. Bañares et al. analysed many of the variables associated with a low-grade coeliac enteropathy diagnosis (patients with suspected CD but without villous atrophy) [21] and developed a scoring system that was able to identify these patients with an area under the curve value of 0.91 [21]. In the present study, we considered an uncertain CD diagnosis if villous atrophy was not present in the duodenal biopsies; in these cases, the performance of an IEL-FC at a baseline work-up endoscopy allowed the clinician to reach a definitive diagnosis for more than 90% of patients.

We are aware of some limitations of our study. First, not all the Catassi and Fasano’s criteria were assessed since clinical and/or histological responses to a GFD were not specifically recorded. However, it is also true that a second histologic evaluation is often dismissed if the symptoms disappear after a GFD assumption. Second, an IEL-FC was performed for the initial CD work-up in some patients who showed villous atrophy in duodenal biopsies. Even if we exclude patients undergoing a GFD, other relevant data associated with villous atrophy, such as angiotensin 2-receptor blockers use or a parasite infection [11,12,13,16], were not registered in our database. Although we believe that an IEL-FC should be considered for inclusion in the diagnostic algorithm of CD, particularly in doubtful cases or patients already on a GFD, the design of our study is not suitable to assess the optimal cut-off for an IEL-FC celiac pattern and further prospective, where validated cohort studies are still needed.

## 6. Conclusions

In summary, the use of the IEL-FC facilitated a diagnosis in a number of patients in our cohort, mainly doubtful cases and patients undergoing a GFD. Further studies are needed to determine under which circumstances the performance of an IEL-FC does not provide any additional value to a CD work-up.

## Figures and Tables

**Figure 1 nutrients-16-01633-f001:**
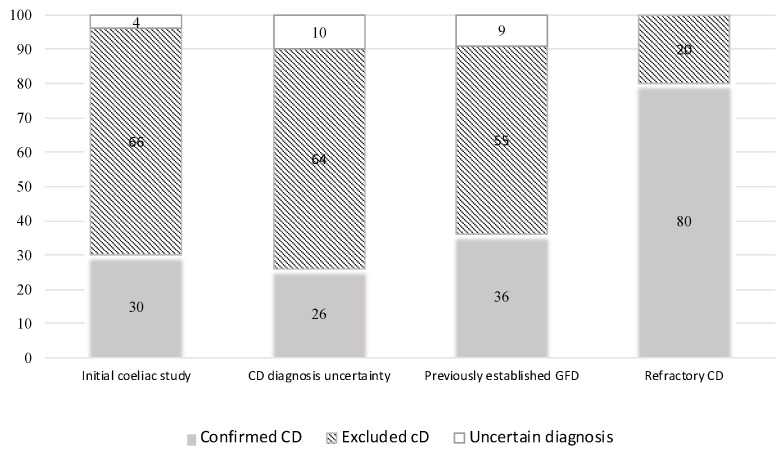
Definitive diagnosis according to the indication of intraepithelial lymphocyte immunophenotyping.

**Figure 2 nutrients-16-01633-f002:**
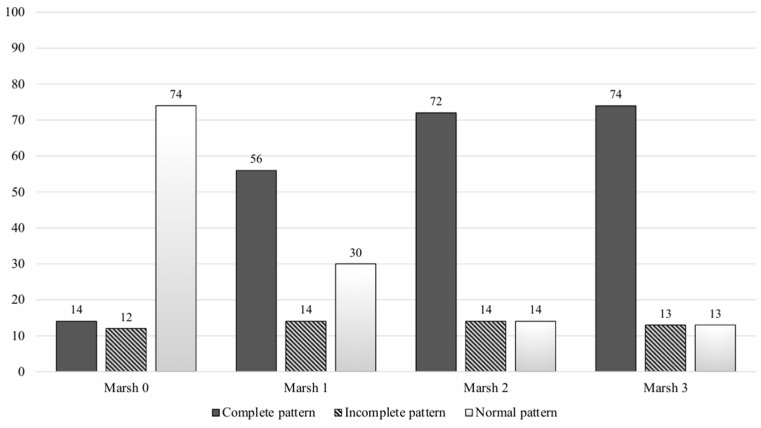
Intraepithelial lymphocyte immunophenotype according to histologic findings.

**Figure 3 nutrients-16-01633-f003:**
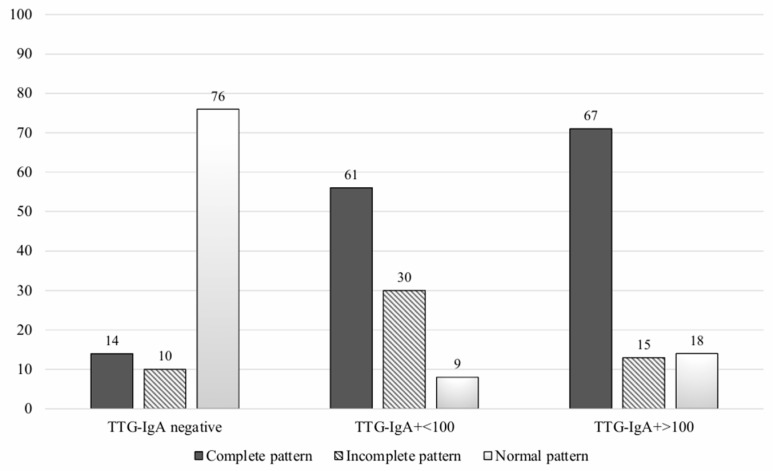
Intraepithelial lymphocyte immunophenotype according to serological testing.

**Table 1 nutrients-16-01633-t001:** Clinical and epidemiological features of the cohort (n = 348).

*Age*, Median, sd	44.76 ± 15.75
***Female gender,* ** *n (%)*	250 (72)
***Familial history of coeliac disease,* ** *n (%)*	64 (18)
***High-risk population,* ** *n (%)*	30 (9)
***Gastrointestinal symptoms,* ** *n (%)*	231 (66)
***Iron deficiency anaemia,* ** *n (%)*	70 (20)
***Liver enzyme alteration,* ** *n (%)*	31 (9)
***Positive genetic study*** *(available in 295), n (%)* - *HLA-DQ2 positive* - *HLA-DQ8 positive*	211 (61) 152 (44) 59 (17)
***Serologic study,*** *n (%)*	
*tissue-transglutaminase IgA antibodies +*	95 (27)
*anti-endomysium IgA antibodies + (available in 273)*	36 (10)
***Histologic results before**intraepithelial lymphocyte immunophenotype*** *(available in 179), n (%)* - *Normal* - *Lymphocytic enteropathy* - *Cryptal hyperplasia* - *Duodenal villous atrophy*	86 (48) 59 (33) 4 (2) 30 (17)

**Table 2 nutrients-16-01633-t002:** Distribution of patients according to the indication of intraepithelial lymphocyte immunophenotyping (n = 348).

Indication for Intraepithelial Lymphocyte Immunophenotyping n (%)		Intraepithelial Lymphocyte Immunophenotype		
		Complete Coeliac Pattern n (%)	Incomplete Coeliac Pattern n (%)	Normal Pattern n (%)
*Initial coeliac disease work-up*	137 (4)	40 (29)	13 (10)	84 (61)
*Uncertain diagnosis of coeliac disease*	112 (32)	26 (23)	18 (16)	68 (61)
*Previously established gluten-free diet*	94 (27)	30 (32)	13 (14)	51 (54)
*Refractory coeliac disease*	5 (1)	3 (60)	2 (40)	0 (0)
*Total*	348	99	46	203

**Table 3 nutrients-16-01633-t003:** Combined results of histology, serology and intraepithelial immunophenotype.

	Marsh 3 on Histology		Marsh 9 ≠ 3 on Histology	
*Immunophenotype*	*Anti-tGT +*	*Anti-tGT −*	*Anti-tGT +*	*Anti-tGT −*
** *Complete CD pattern* **	36	9	25	29
** *Incomplete CD pattern* **	6	3	16	21
** *Normal* **	3	11	9	180
	45	23	50	230

Anti-tGT = antibodies against tissue transglutaminase; CD = coeliac disease.

**Table 4 nutrients-16-01633-t004:** Factors associated with histological findings consistent with coeliac disease (Marsh 3).

	Univariate Analysis	Multivariate Analysis HR (IC 95%)
*Gender*	0.145	
*Gastrointestinal symptoms*	0.141	
*Familial history of coeliac disease*	0.223	
*Other high-risk populations*	0.584	
*Iron deficiency anaemia*	**0.075**	0.30 (0.57–3.21); *p* = 0.498
*Liver enzyme abnormalities*	**0.061**	0.50 (0.46–5.89); *p* = 0.442
*HLA-DQ2+*	**0.006**	0.15 (0.53–2.56); *p* = 0.704
*HLA-DQ8+*	**0.025**	0.98 (0.1–1.42); *p* = 0.148
tissue-transglutaminase IgA antibodies +	**<0.001**	**2.03 (3.35–17.07); *p* < 0.001**
anti-endomysium IgA antibodies +	**<0.001**	0.31 (0.49–3.79); *p* = 0.547

## Data Availability

The dataset is available upon request from the authors.

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
