# Peer review of "The Usefulness of Intraepithelial Lymphocyte Immunophenotype Testing for the Diagnosis of Coeliac Disease in Clinical Practice"

_nutrients, 2024, doi:10.3390/nu16111633_

Round 1

Reviewer 1 Report

Comments and Suggestions for Authors

The paper is very interesting but the conclusions don't permit to insert the dosage of Intraepithelial Duodenal Lymphocytes Flow Citometry in the programm to diagnose Celiac Disease

I ask to the Author a revision of the data because they speak about 348 patients and in the table the results are different

Author Response

We appreciate the reviewers' comments and hope to address all of them.

Reviewer #1

The paper is very interesting but the conclusions don't permit to insert the dosage of Intraepithelial Duodenal Lymphocytes Flow Cytometry in the program to diagnose Celiac Disease

The aim of our study was not to consider the best cut-off for diagnosis but the usefulness of duodenal lymphocyte flow cytometry in the diagnostic algorithm for celiac disease in challenging cases, such as those patients with initial negative serology or those who are on a gluten-free diet before duodenal biopsy, in whom standard diagnosis based on the Catassi and Fasano’s criteria would not be feasible. In this sense, our study provides more data on this matter. Moreover, due to the study retrospective and multicenter design, this conclusion cannot be extrapolated. We have added a sentence in the discussion regarding this issue.

I ask to the Author a revision of the data because they speak about 348 patients and in the table the results are different.

We apologize for the mistakes. In agreement with the reviewer, all presented data have been reviewed and modified accordingly.

Reviewer 2 Report

Comments and Suggestions for Authors

The intent of this manuscript was to establish the clinical utility of the IEL-FC in specific clinical scenarios – coeliac disease (CD) workup, uncertain CD diagnosis, already on GFD before biopsies, and refractory CD. Whilst a series of prior studies support the potential merit of performing IEL-FC in the work-up of CD, unfortunately the data in this study was presented suboptimally and the actual value of the IEL-FC, how it contributes to the final diagnosis, and how it might be deployed in the clinic remains unclear.

Major Issues:

1.     Issues related to data presentation

(i)             All the cohorts were presented as a single cohort despite have very different clinical phenotypes. This approach, including Table 1, is simply not informative and also confusing when you are combining people with uncertain CD diagnoses and refractory CD (which has an entirely different IEL phenotype!) etc all together. The data must be separated by cohort.

(ii)           It remains unclear how the IEL-FC helped rule in or out a diagnosis of CD. Figure 1 is difficult to interpret. How was the IEL-FC result actually used to rule in or out a diagnosis? What about an incomplete coeliac pattern and its interpretation?

“More than 75% of patients with VA showed a complete CD pattern in IEL-FC, whereas 73% of patients with Marsh 0 presented a normal pattern in IEL-FC (p<0.001)” This really doesn’t provide much information on the discriminant value of the IEL-FC in the clinical algorithm.

Distinct from Figure 1, it would be more informative to see the IEL-FC results graphed against diagnosis.

Further, it would be preferable to present sensitivity/specificity/AUC for the performance of the IEL-FC in each patient cohort against the final diagnosis.

(iii)         “Normal IEL-FC was found in 215 patients (62%) allowing us to rule out the diagnosis of CD”. How? If a patient has positive serology, consistent histology and clinical features of coeliac but had normal IEL-FC how is this interpreted?

Similarly, “CD diagnosis was confirmed by a complete coeliac pattern in 108 patients (31%)” (lines 148-149). How did it manage to do this? Or is this statement simply stating that the IEL-FC results were concordant with the final diagnosis?

(iv)          Introduction: “We aim to describe the current use of IEL-FC…and assess its impact on CD diagnosis and the correlation between the cytometric pattern and serological, clinical, and histological criteria.” However, no correlation was provided. This would be useful data to show.

2.     Issues related to the definition of CD

(i)             Introduction: “CD diagnosis is based on the Catassi and Fasano’s criteria, which include….”.  This comment is then referenced by the ESSCD guidelines and BSG guidelines, which is entirely appropriate when referring to diagnostic criteria.  However, Profs Fasano and Catassi are not authors on either of those guidelines. The authors should stick to referencing these guidelines and not referring to Catassi and Fasano.

(ii)           “CD diagnosis is reached when four out of these five criteria are fulfilled or, in the absence of a genetical study, three out of four”. This would not ne accepted consensus criteria. Either use the correct reference (which I presume relates to Catassi/Fasano) or remove this.

(iii)         First sentence in Abstract: “The diagnosis of coeliac disease (CD) in adults is based on clinical, serological, genetic and histological criteria.” Genetic data may help exclude a diagnosis of CD but it is not part of the accepted diagnosis.

(iv)          “Moreover, the diagnostic criteria for CD were modified to include mild histological enteropathy (Marsh 1) [1-6].”  While there may be a trend by some experts to encompass Marsh 1 changes within the spectrum of CD, generally this statement is not correct. In fact, the aforementioned references from ESSCD and BSG explicitly state that Marsh 1 change is not sufficiently specific for CD.

(v)            “The definitive diagnosis (CD confirmed, uncertain or ruled out) was given by the attending physician after IEL immunophenotyping was registered.”

-       Where is this data?

-       How was a definitive diagnosis made?

-       Was it made before or after the IEL-FC data and did it take the IEL-FC data into consideration?

Other Issues

(i)             Results: A table showing indication for IEL-FC, investigation results and final diagnosis should be provided.

(ii)           A single biopsy was collected from D1 and D2.  This seem low to provide enough cells for IEL-FC? How often was it not?

(iii)         Separate to looking simply for the complete or incomplete pattern, did you analyse the percentage cut-offs to outcomes?  Others have shown other cut-offs can work e.g. Garcia-Hoz et al, Intraepithelial Lymphogram in the Diagnosis of Celiac Disease in Adult Patients: A Validation Cohort, Nutrients 2024 used 14% increase in TCRgd IELs and 4% decrease in CD3- IELS.

(iv)          Discussion: “Our study shows that once IEL-FC is introduced in a centre its utilisation spreads rapidly and it is applied to any patient with suspected CD, even though IEL-FC is likely unnecessary for those patients for whom a diagnosis could be reached simply with the “four of five” rule.”

-       What does “utilisation spreads rapidly” mean?

-       If it is sometimes unnecessary, when should the IEL-FC be applied? A recommended clinical algorithm would help provide clarity on its clinical positioning

(v)           A challenge of performing flow cytometry on IELs is ensuring fresh samples and adequate biopsies. A comment on the practicalities of flow cytometry would be of interest – for example, how many biopsies are needed to provide adequate cell numbers for analysis? How long can they sit in normal saline before they need to be processed? Would most hospital departments have the ability to undertake this testing?

(vi)          References incomplete. E.g. Ref 8 – what year? Ref 2; L missing from Lundin.

(vii)        Need to reference a relevant, recent paper: Intraepithelial Lymphogram in the Diagnosis of Celiac Disease in Adult Patients: A Validation Cohort, Nutrients 2024.

(viii)      Line 65 Toll-like receptor – should be T cell receptor

(ix)          Line 69 - use symbol for yd

(x)           Lines 114-115 - include figure of gating strategy or reference previous publication using this.

(xi)          Table 1 – revise histological results section

a.      Are the categories independent or can participants fall under several?

b.     Does each category relate to Marsh 1, 2, and 3 – if so, use this label

c.     Use ‘duodenal villous atrophy’ instead of ‘villous duodenal atrophy’.

(xii)        Line 147 – in text it says normal IEL-FC was found in 215 patients but in Table it says 203.

(xiii)      Figure 1 – suggest it is entirely reworked

Comments on the Quality of English Language

Some minor grammatical issues identified but should be easily addressed.

Author Response

We appreciate the reviewers' comments and hope to address all of them.

Reviewer # 2

Major Issues:

  1. Issues related to data presentation

(i)             All the cohorts were presented as a single cohort despite have very different clinical phenotypes. This approach, including Table 1, is simply not informative and also confusing when you are combining people with uncertain CD diagnoses and refractory CD (which has an entirely different IEL phenotype!) etc all together. The data must be separated by cohort.

We agree with the reviewer that the entire cohort included very different clinical phenotypes. However, these subgroups are clearly defined in the Methods section. Moreover, results regarding intraepithelial immunophenotype and definitive diagnosis are presented according to these clinical phenotypes (Table 2 and Figure 1). This allows readers to compare the outcomes of the technique across different clinical scenarios, which was one of the aims of our study.

(ii)           It remains unclear how the IEL-FC helped rule in or out a diagnosis of CD. Figure 1 is difficult to interpret. How was the IEL-FC result actually used to rule in or out a diagnosis? What about an incomplete coeliac pattern and its interpretation?

In the Methods section, we explained that the attending physician provided the definitive diagnosis (confirmed CD, uncertain CD, or ruled out CD) after registering the IEL immunophenotyping. Additionally, we added a sentence in that section explaining how the incomplete celiac pattern was interpreted and used to rule in or out the diagnosis

“More than 75% of patients with VA showed a complete CD pattern in IEL-FC, whereas 73% of patients with Marsh 0 presented a normal pattern in IEL-FC (p<0.001)” This really doesn’t provide much information on the discriminant value of the IEL-FC in the clinical algorithm.

The reviewer is right in stating that we did not provide much information on the discriminant value of the IEL-FC. In order to solve this issue, we have added a new table (Table 3) showing the combination of histology, serological status and IEL-FC results. Moreover, we have added a paragraph in the Results section to shed more light in this setting.

Distinct from Figure 1, it would be more informative to see the IEL-FC results graphed against diagnosis. Further, it would be preferable to present sensitivity/specificity/AUC for the performance of the IEL-FC in each patient cohort against the final diagnosis. 

As far as the final diagnosis took into account the results of IEL-FC, we think that a graph of IEL-FC results against the final diagnosis would not add any relevant additional information.

(iii)         “Normal IEL-FC was found in 215 patients (62%) allowing us to rule out the diagnosis of CD”. How? If a patient has positive serology, consistent histology and clinical features of coeliac but had normal IEL-FC how is this interpreted?

We thank the reviewer’s comment. In the new Table 3 It is clearly stated that 3 out of 45 patients with Marsh 3 histology and a positive serology showed a normal epithelial immunophenotype. In these cases, a careful evaluation is advisable and, in fact, IEL-FC should not be recommended. Moreover, 89% of normal IEL-FC occurred in patients with a non-definitive histology and negative serology; in these cases, IEL-FC permitted to rule out the diagnosis of CD. The sentence in the results section has been modified accordingly.

Similarly, “CD diagnosis was confirmed by a complete coeliac pattern in 108 patients (31%)” (lines 148-149). How did it manage to do this? Or is this statement simply stating that the IEL-FC results were concordant with the final diagnosis?

We apologize because the numbers were not correct in the text but at the Table. We have modified the data accordingly. Once again, the reviewer is right in suggesting that the information we provided was difficult to interpret. With the new table (Table 3) it is easy to see that although 36 out of 99 patients with a complete coeliac pattern had Marsh 3 histology and positive serology, there were 38 patients with a negative serology and 54 with a non-definitive histology that showed a complete coeliac pattern. All these data are also commented in a new paragraph in the Results section.

(iv)          Introduction: “We aim to describe the current use of IEL-FC…and assess its impact on CD diagnosis and the correlation between the cytometric pattern and serological, clinical, and histological criteria.” However, no correlation was provided. This would be useful data to show.

Accordingly, to the reviewer’s comment, we have modified this sentence. Although it would be very useful to provide the correlation between cytometric and serological, clinical, and histological criteria, we only provide the association analysis between IEL-FC/histology and IEL-FC/serology.

  1. Issues related to the definition of CD

(i)             Introduction: “CD diagnosis is based on the Catassi and Fasano’s criteria, which include….”.  This comment is then referenced by the ESSCD guidelines and BSG guidelines, which is entirely appropriate when referring to diagnostic criteria.  However, Profs Fasano and Catassi are not authors on either of those guidelines. The authors should stick to referencing these guidelines and not referring to Catassi and Fasano.

We appreciate this remark, reference has been changed accordingly (reference #4)

(ii)           “CD diagnosis is reached when four out of these five criteria are fulfilled or, in the absence of a genetical study, three out of four”. This would not ne accepted consensus criteria. Either use the correct reference (which I presume relates to Catassi/Fasano) or remove this.

We appreciate this remark, reference has been changed accordingly (reference #4)

(iii)         First sentence in Abstract: “The diagnosis of coeliac disease (CD) in adults is based on clinical, serological, genetic and histological criteria.” Genetic data may help exclude a diagnosis of CD but it is not part of the accepted diagnosis.

We have modified the abstract in agreement with the reviewer.

(iv)          “Moreover, the diagnostic criteria for CD were modified to include mild histological enteropathy (Marsh 1) [1-6].”  While there may be a trend by some experts to encompass Marsh 1 changes within the spectrum of CD, generally this statement is not correct. In fact, the aforementioned references from ESSCD and BSG explicitly state that Marsh 1 change is not sufficiently specific for CD.

In agreement with the reviewer, this statement was removed from the manuscript.

(v)       “The definitive diagnosis (CD confirmed, uncertain or ruled out) was given by the attending physician after IEL immunophenotyping was registered.” 

-       Where is this data?

Figure 1 show the definitive diagnosis according to the indication for intraepithelial lymphocyte immunophenotyping. We have changed the figure’s title to be clearer and use the same terms as in the methods parts.

-       How was a definitive diagnosis made?

We added a sentence in the Methods section explaining how the incomplete celiac pattern was interpreted and used to rule out the diagnosis.

-       Was it made before or after the IEL-FC data and did it take the IEL-FC data into consideration?

As it is now stated in the Methods section the definitive diagnosis was given by physician after the results of IEL-FC, taking into consideration all the available clinical, histological, serological and immunophenotypic data.

Other Issues

  • Results: A table showing indication for IEL-FC, investigation results and final diagnosis should be provided.

A new Table (Table X) has been made and it can be added according to the editors’ decision.

(ii)           A single biopsy was collected from D1 and D2.  This seem low to provide enough cells for IEL-FC? How often was it not?

 According to our protocol, performing two biopsies (one of D1 and D2) is sufficient to provide enough cells for IEL-FC. Obviously, additional tissue samples were collected for the histological study. We have specified it in the methods section.

(iii)         Separate to looking simply for the complete or incomplete pattern, did you analyse the percentage cut-offs to outcomes?  Others have shown other cut-offs can work e.g. Garcia-Hoz et al, Intraepithelial Lymphogram in the Diagnosis of Celiac Disease in Adult Patients: A Validation Cohort, Nutrients 2024 used 14% increase in TCRgd IELs and 4% decrease in CD3- IELS.

Although the reviewer is right in the interest of assessing different cut-off points, we followed the cut-off proposed by Prof. Fernández-Bañares group. In fact, the technicians who perform the IEL-FC reading learnt it in that center. We have included this as a limitation of our study in the discussion.

(iv)          Discussion: “Our study shows that once IEL-FC is introduced in a centre its utilisation spreads rapidly and it is applied to any patient with suspected CD, even though IEL-FC is likely unnecessary for those patients for whom a diagnosis could be reached simply with the “four of five” rule.”

-       What does “utilisation spreads rapidly” mean?

It means that after introducing IEL-FC in a medical center, its use quickly becomes widespread. We have changed the word to 'widely' to make it more understandable.

-       If it is sometimes unnecessary, when should the IEL-FC be applied? A recommended clinical algorithm would help provide clarity on its clinical positioning

As stated in the manuscript (last paragraph), our data that the use of IEL-FC seems to be particularly useful for doubtful cases and patients already undergoing GFD. However, given the retrospective design of our study and its limitations (also mentioned in the discussion), further studies are needed to determine under which circumstances the performance of IEL-FC does not provide any additional value to CD work-up.

(v)           A challenge of performing flow cytometry on IELs is ensuring fresh samples and adequate biopsies. A comment on the practicalities of flow cytometry would be of interest – for example, how many biopsies are needed to provide adequate cell numbers for analysis? How long can they sit in normal saline before they need to be processed? Would most hospital departments have the ability to undertake this testing?

Although the methodology for IEL-FC is already explained in the methods section, we have added a sentence in the discussion about how this could harm the ability to undertake this testing in different hospitals. However, this study is not focused on the lab methodology but on clinical implementation.

(vi)          References incomplete. E.g. Ref 8 – what year? Ref 2; L missing from Lundin.

We have completed both references in the manuscript.

(vii)        Need to reference a relevant, recent paper: Intraepithelial Lymphogram in the Diagnosis of Celiac Disease in Adult Patients: A Validation Cohort, Nutrients.

The reference has been added (reference number 10).

(viii)      Line 65 Toll-like receptor – should be T cell receptor

We have modified this sentence accordingly.

(ix)          Line 69 - use symbol for yd

We have modified this sentence accordingly.

(x)           Lines 114-115 - include figure of gating strategy or reference previous publication using this.

 We have referenced this in the manuscript.

(xi)          Table 1 – revise histological results section

Tables and manuscript data have been reviewed and modified accordingly.

  1. Are the categories independent or can participants fall under several?

All categories are independent.

  1. Does each category relate to Marsh 1, 2, and 3 – if so, use this label

We have modified the label to Marsh 1,2 and 3 in the whole manuscript.

  1. Use ‘duodenal villous atrophy’ instead of ‘villous duodenal atrophy’. 

We have modified this sentence in the current manuscript.

(xii)        Line 147 – in text it says normal IEL-FC was found in 215 patients but in Table it says 203.

Tables and manuscript data have been reviewed and modified accordingly.

(xiii)      Figure 1 – suggest it is entirely reworked

We have reworked the figure as suggested.

Comments on the Quality of English Language

Some minor grammatical issues identified but should be easily addressed.

We have addressed the English language.

Reviewer 3 Report

Comments and Suggestions for Authors

Quite an interestingly written article about a serious disease called celiac disease (CD). Many factors can make the diagnosis of this disease difficult. A valuable method for diagnosing celiac disease may be the determination of the intraepithelial lymphogram of the duodenum using flow cytometry (IEL-FC), described by the authors. The research group was defined. The most important conclusion that can be drawn from the above work is that the IEL-FC test facilitated the final diagnosis in 93% of patients. A quick diagnosis shortens the diagnostic process and allows for faster implementation of medical procedures. The results were statistically analyzed by performing the Chi2-squared test and the t-Student test. However, I miss the statistically significant differences between the groups indicated in Figure 2. Not much literature. Celiac disease is a commonly known disease, so you can also review the literature and add more items.

Author Response

We appreciate the reviewers' comments and hope to address all of them.

Reviewer #3

Quite an interestingly written article about a serious disease called celiac disease (CD). Many factors can make the diagnosis of this disease difficult. A valuable method for diagnosing celiac disease may be the determination of the intraepithelial lymphogram of the duodenum using flow cytometry (IEL-FC), described by the authors. The research group was defined. The most important conclusion that can be drawn from the above work is that the IEL-FC test facilitated the final diagnosis in 93% of patients. A quick diagnosis shortens the diagnostic process and allows for faster implementation of medical procedures. The results were statistically analyzed by performing the Chi2-squared test and the t-Student test.

However, I miss the statistically significant differences between the groups indicated in Figure 2. Not much literature. Celiac disease is a commonly known disease, so you can also review the literature and add more items.

We thank the reviewer's feedback, we have included and clarified certain concepts and literature regarding celiac disease in the manuscript.